# Vulnerability, Risk and Harm for People Who Use Drugs and Are Engaged in Transactional Sex: Learning for Service Delivery

**DOI:** 10.3390/ijerph19031840

**Published:** 2022-02-06

**Authors:** Catriona Matheson, Lesley Bon, Louise Bowman, Adrienne Hannah, Katy MacLeod

**Affiliations:** 1Independent Research Consultant, Aberdeen AB10 6RT, UK; 2Scottish Drugs Forum, Glasgow G1 3LN, UK; lesleyb@sdf.org.uk (L.B.); louise@sdf.org.uk (L.B.); adrienne@sdf.org.uk (A.H.); katy@sdf.org.uk (K.M.)

**Keywords:** transactional sex, drug use, women, services, drug related harm, trauma

## Abstract

Against a backdrop of high levels of drug related harms in Scotland, this research aimed to inform training development to aid the upskilling of the workforce to support people who use drugs and are involved in transactional sex. Sixteen qualitative interviews were conducted with participants recruited across four health board areas and a range of services covering sexual health and/or harm reduction. Interviews explored initiation, frequency, and the nature of transactional sex; drug use, and experience of services. A thematic analysis was undertaken. Vulnerability was a key emerging theme when discussing drug use, sexual behavior, and service use. Vulnerability increased risk of a range of harms from sexual assault to injecting harms and overdose. Participants were often, but not always, introduced to drugs and transactional sex by someone who had power over them or was more experienced in drug use and/or sex. High risk drug use was evident in terms of polydrug use, consuming large quantities of drugs, sharing crack pipes/injecting equipment. There is a need for information and services to reduce the risk of harm from drugs, sexual violence, and unprotected sex as well as non-judgmental, trauma aware services. The workforce needs to be skilled to recognize and manage these potentially complex syndemics.

## 1. Introduction

### 1.1. Stigma and Health Harms

People who use drugs are subject to stigma, self-stigma, and discrimination [1]. Stigma is described as “a long-lasting mark of social disgrace that has a profound effect on interactions between the stigmatized and the unstigmatized” and self-stigma occurs when people “accept the social meaning of his/her stigma and that stigma must be central to a person’s sense of self” [1]. Stigmatization can come from health professionals [2]. This is increased for women [3] and further magnified for people who may be involved in transactional sex [4]. This is often, but not exclusively, women. Transactional sex is a broad term that encompasses the selling or exchange of sex for drugs, gifts, money or other services e.g., somewhere to stay. In the interests of inclusion we have adopted the broader concept of transactional sex.

In addition to stigma, people involved in transactional sex who use drugs may experience a range of challenges including exploitation, imprisonment, and traumatic events such as violence and child removal which have an impact on risk taking behaviors relating to drug use and sex. These challenges may affect engagement with support services including the way they engage and level of engagement. There is local anecdotal evidence of women who use drugs describing feeling stigmatized and not welcome at sexual health services. This is worse if they are also selling sex (SDF personal communication). This was also evident in international research [4].

The risks of injecting drug use are well documented and researched [5]. Blood borne virus transmission, injection injuries, and overdose are the most notable. For people who inject drugs who are also involved in transactional sex there is the added risk of sexually transmitted infections and violence [6]. There is evidence globally that sexual health services may focus on men who have sex with men (MSM) and the general population whilst vulnerable female groups such as those involved in transactional sex are inadvertently excluded [7].

There is a considerable body of international literature around women, drugs, and HIV risk that has shaped current service provision. Globally women who use drugs have higher rates of HIV than men who use drugs [8]. However there is also recognition that the range of services (opiate substitution treatment, needle syringe provision, HIV testing antiretroviral treatment) are not yet optimally provided to meet needs and a UNODC consensus statement noted the importance of non-judgmental and tailored services [8]. However HIV is not the most problematic issue facing people who use drugs in Scotland and a broader syndemic approach is required to account for the multiple health and social challenges.

Aldridge et al. measured the health inequalities faced by excluded health populations which include sex workers, homeless populations, prison populations and people who use substances (of which there is considerable overlap) [9]. Disease prevalence was consistently high in these groups for infectious diseases (hepatitis C and B, tuberculosis) and mental health. Raised levels of cardiovascular and respiratory disease were evident. Overall standardized mortality ratios were 8–12 times higher than the general population [6]. Tweed et al. extended this analysis by looking at combinations of exclusion health categories but found few studies had considered the health of sex workers in combination with other exclusion categories apart from substance use (others being homeless, prison populations) [10]. This is a clear evidence gap.

### 1.2. The Scottish Context

This research was conducted in Scotland which has the unenviable record of having the highest level of drug related deaths (DRD) in Europe (3.5 times higher than the rest of the UK) [11]. Whilst the majority of DRD are still in men, the rate of increase of DRD in women, particularly in older age groups is notable. Explanations have been explored in depth and include an ageing cohort, high risk drug use, co-morbidities, parenting roles and child removal and social security changes [12].

Scotland has a national needle exchange surveillance initiative in which data is collected from people using needle exchange services across Scotland allowing trends to be documented [13]. The latest report found participants were 73% male (similar to 71% male in prevalence estimates [14]). HCV antibodies were present in 57% and 2% for HIV. Heroin continues to be the main drug injected. However, there was evidence of increased powder cocaine injecting. Severe soft tissue infections were reported to be up by 20% for 2017–2018 [13]. This demonstrates the harms experienced by Scottish injecting drug users. Unfortunately, analysis was not presented by gender.

Partly in line with previous research recommendations [15], there has been some movement towards a human rights based approach to service delivery as guided by the revised drug strategy ‘Rights, Respect and Recovery’ [16]. However this strategy falls short of integrating gender equality and women focused services, as recommended elsewhere [15]. Although such services exist, it is not the norm. In response to the rise in DRD in women the Scottish Drug Deaths Taskforce (DDTF) initiated a review group to ensure the risks of DRD for women were considered. Recommendations highlighted the need for ongoing gendered awareness [17].

Given the clear risks of transactional sex and injecting drug use, and the very evident and documented high levels of a range of harms in Scotland, it is surprising how little research has been conducted in this high risk, predominately female group. Tweed, in her analysis of increased DRD in women, noted this lack of Scottish research [12].

The Scottish Drugs Forum (SDF) is a national, membership-based resource that delivers training (amongst other activities). In line with the absence of specific training on the harms associated with drug use for those involved in transactional sex, SDF was commissioned to undertake this research to underpin an evidence-based training resource. This research aimed to identify information, support and treatment needs associated with drug use, sexual health and BBV risk taking behaviours of people involved in transactional sex. This would enable the delivery of a gender aware, rights-based approach to services to reduce risk to this highly stigmatised and under-served group.

## 2. Materials and Methods

A qualitative approach was used with semi-structured interviews to collect data. Data collection was conducted pre-COVID from February 2019 until August 2019.

### 2.1. Sampling and Participation

Four health board areas in Scotland (Greater Glasgow and Clyde (GGC), Lothian, Fife and Grampian) were purposively selected for participation, to provide a range of geographical areas that included large conurbations (Glasgow, Edinburgh, Aberdeen) as well as rural areas and smaller towns in Grampian and Fife. Sexual and reproductive health clinics, injecting equipment providers (IEPs) who distribute equipment and paraphernalia and/or services that specialize in transactional sex were asked to display posters and host a researcher on a set day/days to be available for interviews on site. Inclusion and exclusion criteria for interview participation were: people (women or men) involved in transactional sex who use drugs, 18 years or over, able to provide informed consent and live in Scotland. Target recruitment was 20 participants, or until data saturation was reached.

Those meeting the inclusion criteria were given a participant information leaflet by the service staff or forwarded by the researcher if from an independent enquiry. They were told that the researcher would be present in the service on a specific day(s) to undertake interviews. Those expressing an interest were given an appointment date and time. Flexibility was important and researchers were available to undertake interviews at short notice, if requested by participants.

In addition to face-to-face interviews there was an option for a telephone interview. Posters, cards, and fliers were displayed and distributed widely through services. These gave a phone number to make contact (call or message). A web site link provided more information on the study. Social media was used to raise awareness.

### 2.2. Interview Content and Conduct

Interviews were conducted by two trained SDF researchers with oversight from an experienced researcher. A semi-structured topic guide was developed with input from an advisory group and informed by the literature. Topics covered were: first experiences of drug use (setting, who with, drugs used and route of administration and frequency of use), first experience of transactional sex; frequency and type of transactional sex; sources of information on drugs; use of support services or treatment for drug use; and experiences of support or treatment around drug use and/or sexual health services and service needs. Whilst there were set topics and opening questions that had to be asked of all participants, these did not have to be in the order they appeared on the topic guide and follow up questions were used to draw out further detail. This was intended to encourage a more conversational approach to the interview. Interviewers were trained in this approach.

Interviews were digitally audio-recorded with participants’ consent. Brief descriptive data (age, gender, sexual orientation) were collected on each participant to provide the context for qualitative responses. Interviews were undertaken face-to-face or by telephone. Informed consent was recorded verbally allowing participants to remain anonymous. Interviews lasted between 30 and 60 min.

All data were fully transcribed by an experienced transcriber. The first few interviews for each interviewer were checked by the lead author and feedback provided to ensure a consistent approach was being used between researchers. A thematic analysis was undertaken [18]. Initially an analytical framework was developed by the research team who read and coded a sample of interviews. These were collectively reviewed and organized into an analytical framework. The thematic framework was then systematically applied to all interviews. Broad themes generally related to topics explored in the topic guide: drugs used, experience of transactional sex, exposure to harms, sources of information, support, and treatment. Sub themes related to the range of experience and views within themes. Findings are presented using this format as this was considered the most accessible for a mixed practitioner and clinical audience. Cross cutting themes emerged at analytical level and were added to the framework.

## 3. Results

### 3.1. Participants

A total of 15 women and 1 man were interviewed aged from 24–51 years with a mean age of 37.6 years (standard deviation 8.05). Eight identified as heterosexual, seven considered themselves bisexual, and one identified as a gay man. The majority were Scottish with one person being Eastern European. (Table 1) The self-identity of women, and the terminology they used when referring to themselves was interesting since participants often seemed surprised to be asked. They needed some prompting with suggestions. They generally settled on: working girl, escort, or prostitute. The most frequent term used for men they had sex with was ‘punters’, but ‘client’ was also used.

Participants worked in a range of settings including working the streets, via brothels, escort agencies, saunas or online via direct personal accounts. The number of customers seen in a working time period varied considerably from 1 or 2 clients a day (and not every day) to 12. Some participants worked for a few days every week whilst one worked two or three times a month.

Whilst findings are presented according to the broad themes (drugs used, experience of transactional sex, harms-drugs and sexual health, information needs and support needs) which map onto the topics explored, there were cross cutting themes of shame, coercion and vulnerability. The strongest emerging cross cutting theme across all participant interviews which was that of vulnerability. Some were vulnerable since they were young when first introduced to drugs and/or transactional sex. Some had challenging early life circumstances including being taken into the care system. This increased their exposure to people who would take advantage of their vulnerabilities as well as to other women and girls who were already involved in transactional sex. This theme of vulnerability is highlighted throughout.

Note that when quotes are used in this section R refers to the researcher and P to the participant. Participant codes are included in brackets at the end of each quotation.

### 3.2. Drugs Used

There was a broad range of drugs used and poly-drug use was commonly described. Many participants started their drug use with heroin, often not knowing what they were using. This made participants vulnerable to the effects of the drugs as well as to what they were exposed to whilst under the influence of substances. Drugs, and alcohol, could be used deliberately to make it easier to take part in sexual acts and to block out feelings of shame:


*For my first time, what it was, right, is they told me obviously about heroin, and that, and I was a wee bit obviously wary, and plus they were telling me that I had a client and that coming, she was there obviously in the same house at that time, so I was a wee bit nervous and that, scared, a bit dirty, I think then, so she said here, take this, this will make, calm you down and that, and it did, you know, after I was, I did the deed, I did feel dirty and that, and then that kind of got me into the situation that I needed the drugs to obviously get me out in that zone*
(326)

Most but not all participants (one person had a codeine dependence) used heroin at some point in their life. A few were on a methadone prescription and not using heroin on top. A few people described smoking heroin and not injecting. However there was widespread experience of heroin injecting across the group. Initiation into injecting was usually described as being facilitated by other people including boyfriends and friends.


*“… it was one of my, my friends, they, it was one of their contacts, she gave me the heroin, she was in the same house, and they seen I was really nervous [about having sex], she obviously made up the kit, that was heroin, and gave me it, injected me, because I didn’t have a scooby [clue] what to do, and then the guy come in, we went to the room, and just got on with it.”*
(326)

This was also an initiation into transactional sex, the drugs enabling the sex to take place.

High risk poly drug use was described by some participants:


*“Yeah, I use coke, crack, I use heroin, valium, gabapentin, anything, I like to be out of my face, when I’m in that mood.”*
(326)

‘Street’ benzodiazepines including ‘Xanax’ and ‘Valium’, also described as ‘benzos’, were common and described as being cheap, leading to high levels of use. High risk use was also evident in terms of the amount of benzodiazepines people would take:


*“Vallies, god I could take 20, 30 in the one day I would say”. *
(326)

Another participant described how someone was given drugs for free. It was unclear from the interview why she was targeted in this way.


*“Well they were, they [tablets] were getting posted through my door, so I wasn’t even having to pay for them, and I’m like, oh no, here we go, but I don’t, because of my size and build, it doesn’t take a lot, but you get what, 15 for a tenner or something, …. I wouldn’t be able to remember for days.” *
(344)

Some participants had strong preferences for particular drugs while others expressed specific dislikes for certain drugs including benzodiazepines, gabapentin and pregabalin. One participant described an unpleasant experience using ketamine including memory loss and black outs:


*“Oh yeah, I did not like that, he had ketamine in the house, and he used to get it from this girl, and I tried, I tried some of it, and I’d been awake for a few days, and I, it was like a, I had blackouts, I don’t know what happened, I don’t, I mean I know I was there, I know I wasn’t passed out or anything, but it’s just like pure gaps, and I wasn’t really drinking or anything, so it was really bizarre, they just blocked out my memory, and I know that there was nothing really that happened, it was just the two of them chatting, but that was not a nice experience, too trippy, too mad”*
(324)

Crack cocaine use was described as becoming more popular and several participants described using it:


*“Yeah, it’s so prevalent now it’s overtaken smack, definitely.” *
(327)

Initiation in crack use came from a range of people including clients as this participant described:


*“me and this guy, like we just had this great chemistry, it was just everything that I wanted, he was everything I bloody wanted, right, … I never stopped to question that the first couple of times we’d met, he just stayed really quiet, he was studying me, ….I didn’t see that, I was being manipulated, my god he was good…, and then he, he said right, you’re not smoking this thing properly, you’ve got to breathe it in, and then hold it in, and Jesus, my god, I have never felt anything so amazing my life….and that’s when I knew, right, I, I need to marry this man, I want to smoke this stuff every day”. *
(324)

### 3.3. Experience of Transactional Sex

All participants were asked about their first experience of transactional sex. The age at which this took place was very variable from 14 years (one person who had been in care) to 34 years. Some had been groomed, for example the 14-year-old whose ‘boyfriend’ was 24 years old and manipulated her into drug use as a means of controlling her, then led her into selling sex for money for drugs:


*“I was in care, and my boyfriend was 24, and he basically pimped me out, so and got me onto heroin, by telling me it was cocaine, and gave me it for 6 months, then told me what it was, and I was addicted to it by that point, so I had no [choice] to take it.” *
(327)

Another participant had also been ‘groomed’ by an older man she met through a support service:


*“and I met one older guy from the R******* Hub, but because he was from R******** Hub, I thought he’s very nice, he was very nice person, he was 50 years old, but after he started to buy me presents, clothes, I never, but with him I never asked for his help, but with him it was really, I didn’t want to use even, he, somehow, but he was buying me food, when I finally found hostel, he was knocking my door, like too much, I don’t know, I was a little bit scared and then he started to touch me as well, but because he was older, with him it ended up with police and I didn’t call, because nurses saw how he was hugging me sometimes…” *
(346)

Other people found themselves involved from a place of vulnerability which included desperation for money, not wanting to be caught shoplifting or selling drugs as they might end up in prison and they rationalized that selling sex was preferable. Other vulnerabilities came from being made homeless or having benefits stopped.


*“my benefits had been stopped as well, I was on Universal Credit, so I had no money for eight weeks and that’s kind of how I got into doing it” *
(344)

One participant described the women’s hostel she lived in whilst homeless regularly attracted men looking for sex. Two participants described bereavement and mental health as being key factors in their involvement in transactional sex. A few participants started to work through escort agencies or brothels. A couple of participants described mental health and traumatic experiences as being a key factor in their involvement in transactional sex:


*“R:* 
*So are you okay with me asking you the question about the first time you sold sex, how old were you?*
*P:* 
*I was 24.*
*R:* 
*24, and what was happening at your life at that time that led you there?*
*P:* 
*I lost my partner.*
*R:* 
*Right, and was that through drug death or?*
*P:* 
*Aye.*
*R:* 
*And what happened as a result of that then?*
*P:* 
*I just gave up”*

(326)

One women was 34 when she started an online site for selling sex but she had a troubled past with mental health problems:


*“I had post-natal depression, and post-natal psychosis, … oh loads of shit happened, and my daughter ended up going to stay with her dad and I’d been off it for 13 years, and I packed a bag and buggered off to [place], and ended up meeting a lassie, and got introduced to [name].*
*R:* 
*And did she set up the site for you and stuff like that, or did she just show you how to?*
*P:* 
*She just showed me how, and I did it, I took it from there myself.”*

(323)

Several participants described their shame after the first time they were involved in transactional sex, others described blocking out the experience as a way of coping.


*“[after] the first punter, I got into a hackney, went right home to *** Street, and greeted [cried]” *
(334)

### 3.4. Drug-Related Harms

A range of harms associated with drug use and injecting were explored. Participants were specifically asked about equipment sharing during drug use. All of those who had or were currently still injecting said they had previously shared injecting equipment. None referred to recent injecting and this might reflect their generally older age and long history of drug use but might also be that they did not want to admit to this. There was clearly an understanding that sharing equipment put them at risk of blood borne virus transmission.

At least five participants had had hepatitis C, and one participant had had hepatitis B, all from sharing injecting equipment. Sharing was generally carried out in the context of a partner relationship but also with other drug using acquaintances, particularly when ‘desperate’:


*“I have done that in the past, aye, I’ve used other people’s pins, I’ve used pots, I’ve been desperate and shared, so there is a chance of having Hep C there, aye, I need to get tested again, I think actually, because I’m not sure if I’ve got it or not, you know how your antibodies fight it and all that.” *
(344)

People who initiated participants into injecting were not necessarily skilled. Indeed, scarring was described by a number of participants and one woman had a serious injecting injury that required hospitalization. Several other participants described having injecting injuries such as abscesses, collapsed veins and infections.


*“I was using a lot of old equipment and I didn’t know where to get [it], and I was ordering online, but I had a few of syringes, and I, I have scars on my hands now, and my veins are really bad.” *
(346)

This woman had added risk from injecting legal highs and not being as readily aware of sources of equipment.

Sharing of crack pipes was described by a couple of participants, not due to the fact that they shared themselves but since they had seen the consequences in skin conditions and possible skin infections:


*“They share pipes…Aye, they’ve all got scabs all over their face and aye, so it’s horrible.” *
(325)

### 3.5. Sexual Health Harms

Participants were asked about condom use and contraception as well as testing for sexually transmitted infections through sexual health check-ups. Intermittent condom use was described. Most said they used condoms but not with every client and not all of the time. For example some clients would exert pressure (coercion) by saying they were allergic to latex. Some women described clients using force and violence in order to avoid condom use. One particularly candid participant openly revealed incidences of rape when discussing condom use:


*“R:* 
*So did you, do you always practice safe sex?*
*P:* 
*No, not always, no.*
*R:* 
*Was that an agreement you came with, with people, or was that just, was that like?*
*P:* 
*No, there was a couple of times I was raped, and my first experience was in a sauna and the man raped me anally, and that was really quite a hard experience to go through, but normally,… what was the question again?*
*R:* 
*It was about safe sex, about condom use really.*
*P:* 
*No, I don’t, I don’t always use a condom, because they’ll sometimes ask you, oh I’ll pay a bit extra if you don’t use a condom, and then I’ll judge it by how they are, their appearance and stuff like that, and then just to get my drugs, I would go for it, you know what I mean, and it was more money for drugs.” *

(327)

This participant was also very open in sharing that she would sometimes forgo condom use to be paid extra. The money being used to buy drugs. Her drug dependence making her vulnerable to another layer of sexual risk taking behavior on top of her drug use. Whilst there was some baseline knowledge about the importance of condom use to protect against HIV, Hepatitis C, and unwanted pregnancy, there seemed to be less understanding of the need to have regular smear tests as well as being tested for other STIs:
*“I’ve got, only had so many smear tests in my full life, I’m bad for no going to them, aye.” *(334)

### 3.6. Information Needs

There was a specific need for information on the effects of certain drugs, e.g., on cocaine and the effects on the heart:


*“I asked about cocaine, because I feel that my heart is weird, I was scared and I was told that its better never to inject cocaine, I was not doing that after, because yeah, I was told its better not to, even small amounts.” *
(346)

There was a clear desire for information on drugs expressed by a few participants. Some, as in the example below, were very interested to learn about substances and the effects. The internet was used by some as a source of information. One participant described her thirst for knowledge:


*“…when I try a drug, right, I get quite obsessed about it, I want to know all about it, I want to know its story, it’s like I want to intimately become, meet it….Yeah like I study it, how does it work and so the Seroquel you know, I looked it up and I was like, did you know that this was used recreationally in like the big parties in [city] and things, and they called them Susie Qs.” *
(324)

Information on good injecting practice was also needed. Several participants had relied entirely on the skills of another person, either a friend or partner, to inject them, at least initially.


*“It was a partner I was with, he would inject me and what a mess he would make of me. Looking back I think ‘oh my god’, but that’s why I won’t inject anyone because somebody was at my house not that long ago asking me [to inject for them].” *
(344)

For this particular participant there had been considerable adverse consequences of poor injecting technique as she ended up in hospital with a severe wound infection.

Some used pharmacies for information, picking up the leaflets that might be available. One person mentioned that information about safer injecting and drug taking often came from older women involved in transactional sex. Several participants referred to the need for information on how to manage behavioral issues in others. Some people were aware of their vulnerabilities, or of those in similar positions. One referred to receiving unwanted attention from an older man but not knowing how to manage this. Another was aware of a girl who was struggling to get out of an association with a dangerous and abusive group of people:


*“she’d fell into a gang down there, and she was, I think she was gang raped by them and they were threatening her, that she couldn’t tell anyone, and she didn’t trust anyone, they were telling her that she knew they, they knew people in the police and stuff.”*
(324)

Thus, some expressed needs were for skills as well as information.

### 3.7. Service Needs

The importance of a positive relationship with staff in services was mentioned by several participants. Changes in staff could undermine the development of a meaningful, therapeutic relationship. When asked specifically to describe the most important thing about training staff to work with people involved in transactional sex, this participant expanded on the importance of maintaining reliable, meaningful relationships:


*“Do you know what, listen, and do what you say you’re going to do, it’s two so simple things, right, and keep in touch with the person, don’t just say you’re going to do something and then the person never hears from you again, because then they lose trust in everybody,… the only people we don’t want to lose trust with is this lot, so that would be my main, main thing, is if you promise a woman, you’re going to do something, you do it, and you do it the next day, or as soon as you can, but you give them a timescale.” *
(333)

The importance of feeling listened to, in a non-judgmental manner, was repeatedly highlighted:


*“Yeah, well I was just saying that this is actually the first time that I feel like they have listened to me, because I told them that I, I only wanted 30mL [methadone], I was only using a tenner bag a day, and I thought 30mL would cover it, and the worker is actually listening to me this time, so I feel like I’m a bit more in control about it, so that helps.” *
(344)

This related to underlying concepts of stigmatization in which women feel they are being judged by those that they interact with in services.

Finally people involved in transactional sex often work all night or late into the night and so attending appointments early in the day was very difficult and later appointments or open clinics were noted as possible solutions.

## 4. Discussion

### 4.1. Key Findings

Vulnerability was a key emerging theme when discussing drug use, sexual behavior, and service use. This vulnerability increased risk of a range of harms from sexual assault to injecting harm and overdose. Women were often introduced to drugs by someone who had some sort of power over them or someone who was more experienced in drug use. This might be another women. High risk drug use was evident in terms of polydrug use, consuming large numbers of tablets (presumed to be benzodiazepines by the consumer), sharing crack pipes/injecting equipment and lack of awareness of safe injecting practice, particularly in the early phase of drug use.

### 4.2. Discussion of Findings

A range of drugs are used and taken in a range of ways and both drugs and alcohol are used to make it easier to have transactional sex. Initiation into drug use could be through other women. This finding concurs with that of Tuchman who noted (in a US sample) that other women are often involved in the transition to injecting drug use [19]. Injecting related harm in the form of abscesses as well as transmission of BBV was very evident, in line with large scale surveillance data [13]. High risk drug use in terms of polydrug use was evident. Street benzodiazepines and cocaine use including injecting cocaine was mentioned. This evidence concurs with the picture emerging from the high level of drug related deaths in Scotland [11].

Vulnerability in the women in this study, whether due to being in the care system and not having adequate support, or partnerships with a power imbalance was key to involvement in transactional sex. Involvement in transactional sex and drug use combined increased vulnerability to rape and sexual assault and reference to such traumatic incidents was widespread amongst all of the respondents. Several participants had been raped–some multiple times. Trauma could also stem from the loss of a partner to overdose, or separation from children. The lack of reference to meaningful relationships in people’s lives was notable. There was also considerable shame associated with transactional sex, linking to the concept of self-stigmatization. The sum of all of these very challenging issues is that women involved in transactional sex can be distrustful and harder to engage in services than other service users.

Times of trauma, whether due to poor mental health, loss of a partner, being made homeless and losing benefits are key points when women who use drugs consider involvement in transactional sex. Removal of children can make women vulnerable to exploitation [12]. However a conscious decision is sometimes made around how to make money and transactional sex can be chosen over other activities.

These interviews were conducted pre-COVID. COVID measures reduced face to face service delivery across drug and sexual health services with reliance on telephone and digital contact. This could have impacted the relationships with the service provider, especially for people with limited access to technology. There is also some evidence that sexual behavior will have changed during lockdowns [20], which could have impacted this group.

### 4.3. Information and Service Needs

This trauma background will have a knock-on impact for service providers as it might take time to build trusting relationships. There is already an awareness of the need for trauma-informed service in the drugs field [16,20] but this should be extended to sexual health services. Building relationships on these relatively simple information sharing grounds may pave the way to address the more challenging underlying issues around shame, trauma and sexual assault.

There was a perceived lack of sources of general advice at the point of need which left people vulnerable to the harms of initiation of drug use as well as the harms of poor injecting technique and BBV transmission. Women stressed the importance of continuity of staff, of having trusting, non-judgmental, consistent relationships, with an understanding from staff around the need to sell sex. Interestingly women did not use the term stigma often but instead used the terminology of feeling ‘judged’ which is aligned to feeling stigmatized. Many also described feelings of shame which align to a self-stigmatizing identity.

Several information needs were identified with regards to drugs including effects of drugs, safe injecting practice, managing difficult behavior in others including clients who might take advantage of a person’s vulnerability. Sexual health information needs included testing for STIs and BBVs, condom use, smear tests. Intermittent condom use was evident in our sample. This might be linked to increased money but also to client resistance (condom coercion) and sexual violence. Power imbalances are created due to the vulnerable nature of this group. Research from the U.S. into condom coercion in clients and non-paying partners and intermittent condom use (ICU) found that recent ICU was evident in a significant proportion of female sex workers and this was associated with client intoxication, which can in turn be associated with client violence. There was similar experience of ICU with non-paying partners which was significantly associated with being in a relationship, intoxication at sex, and police violence [21]. Our findings are consistent with this research (with the exception of experience of previous police violence).

There was often some vagueness when asked if and when the women had had a sexual health check-up. Women involved in transactional sex may be unlikely to attend for regular smear tests therefore if someone does attends for a check-up it can be indicative of other potential risk factors, such as being triggered by unprotected sex which in turn may be traumatic in nature again underlining the need for trauma awareness in sexual health services.

Research covering women who use substances and experience domestic violence highlighted that women need to decide which service they wanted to seek– substance use support or domestic violence support [22]. There is a parallel in this study with access to sexual health services, rape support, mental health services and substance use services. People involved in transactional sex might have to choose which of their multiple support needs to address first. Indeed this leads us to consider the syndemics of blood borne virus, sexually transmitted disease, substance use and mental health problems in this group. This lead us to conclude that there needs to be a heightened awareness of the potential for associated drug use harm and sexual health related harm across all of these services as well as the skills to manage such vulnerabilities appropriately without stigma. This will be challenging given the generally stigmatized attitude to people who use drugs across health professionals [2]. With this in mind, and alongside the prematurely ageing cohort of people who use drugs in Scotland and all of the additional morbidity (respiratory and cardiovascular disease) [8] it is essential that a system wide upskilling and destigmatizing of the workforce is undertaken to promote meaningful engagement with services and reduce morbidity and mortality.

### 4.4. Strengths and Limitations of the Study

This was a small study with 16 participants. The sample had good geographical coverage across Scotland and were drawn from a range of services. However, there were key emerging themes across participants and data saturation was generally achieved with women participants. There was the potential for selection bias in that those more able and more engaged with services will have been more likely to participate. Although we tried to broaden inclusion with posters and social media, the most isolated and not digitally engaged may not have participated. Recall bias is always a potential consideration when asking about events in the past. Significant events such as first drug use and first transactional sex experience were generally recalled easily but it is not possible to determine accuracy. One underexplored area was the comparisons or contrasts between men and women who use drugs and their involvement in transactional sex as only one man was interviewed. Further research to explore the risks for men involved in transactional sex is recommended. Involvement of affected women in service design and delivery, as recommended by Tuchmann [19] and the Scottish DDTF [17], is strongly recommended. The development of interventions that address the syndemics of conditions associated with drug use and transactional sex would be a positive, targeted step forward.

## 5. Conclusions

People involved in transactional sex and drug use are often introduced to either or both of these as a result of vulnerabilities due to their circumstances or health. They have often experienced various combinations of drug related harm and sexual violence. There is a need for more information for people involved in transactional sex on health and reducing risk of harm to people as well as non-judgmental, trauma aware services. However, the workforce needs to be skilled to recognize and manage these potentially complex syndemics from a trauma aware stance.

## Figures and Tables

**Table 1 ijerph-19-01840-t001:** Participant descriptors.

Participant	Area	Age	Ethnicity	Gender	Sexuality
1	Fife	26	white Scottish	female	bisexual
2	Fife	36	white Scottish	female	bisexual
3	Fife	40	white Scottish	female	bisexual
4	GGC	31	white Scottish	female	straight
5	Fife	43	white Scottish	female	straight
6	Fife	32	white Scottish	female	bisexual
7	Lothian	37	white Scottish	female	straight
8	GGC	26	white Scottish	female	straight
9	Lothian	51	white Scottish	female	straight
10	GGC	43	white Scottish	female	straight
11	GGC	44	white Scottish	female	straight
12	Lothian	40	white Scottish	male	gay
13	Fife	49	white British	female	whatever goes
14	GGC	42	white Scottish	female	straight
15	Fife	37	white Scottish	female	bisexual
16	Grampian	24	white Latvian	female	bisexual

Note: participant codes are not included here as this information combined with what people have said could make some individuals identifiable.

## Data Availability

Interview data to support findings are available on request.

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
