# Peer review of "Vulnerability, Risk and Harm for People Who Use Drugs and Are Engaged in Transactional Sex: Learning for Service Delivery"

_ijerph, 2022, doi:10.3390/ijerph19031840_

Round 1
Reviewer 1 Report
This is a very well written qualitative study.
Line 59 should probably have (DRD) after drug-related deaths, so we know what DRD stands for later on. Same for SDF on line 84.
Did the bisexual participants (women) engage more clients on average?
Please provide mean and SD for age.
Please discuss/speculate on how the results might differ if they had been done during the middle of the COVID-19 epidemic so readers can better apply the results to their current work with similar participants.
Reviewer 2 Report
The paper addresses an important topic, but the paper suffers from several shortcomings.
First, in the HIV literature, the study of drug use among both males and female sex work populations has been extensive. And this includes using qualitative methodologies in their design. For this paper to have credibility, this literature must be reviewed and incorporated to justify the current study.
Second, a more convincing description of how the interview guide for this study follows the protocol principles used in qualitative methodological is needed. It appears that the questions are set and don't allow participants to open up the discussion as they see fit so that the answers are not simply a response to predetermined questions from the researchers. This questions whether the emerging themes were imposed by the researchers.
Third, it is not clear why there is the inclusion of a male participant given the focus on women? What is the rationale for this decision?
I am not convinced that this study adds anything new to the existing literature. This severe limitation needs to be addressed.
Reviewer 3 Report
Introduction:
The authors should define stigma, and self-stigma in the introduction of the paper.
Please put citations on “The risks of injecting drug use are well documented and researched” page 1, line 40.
Methods:
There are no tables to describe the demographics of those who participated in the study. What is the average age? How many men or women participated in study? What were some of the sub themes? Maybe the authors should include tables so the readers can have a full picture?
Results:
The authors talked about stigma being an issue in the introduction, but it does not seem it was an issue from the participants. Please clarify? If stigma was not discussed by the participants or researchers, why or why not?
Reviewer 4 Report
Overall, this is an interesting study dealing with topic with little information. However, the authors should address important issues to improve the quality of the study.
Introduction
-Lines 49-50. Disease prevalence was higher for whom?
Methods
-Regarding sampling, it seems like the selection of the sample in relation to areas has been at convenience, which implies the issue of a selection bias that can lead to a bias result of your study.
-Target recruitment was 20 participants. Why? Did you conduct a priori power size calculation?
-It is recommendable to provide the participation rate. Too large losses could lead to another selection bias.
-Two independent reviewers conducted the interviews, but was there any kappa coefficient for controlling the agreement?
-There is no statistical part in the methods. Neither descriptive nor analytic.
Results
-Overall, the authors have included subjective comments and observations that does not correspond with a Results section. For instance, lines 160.165. It seems a discussion.
-I do not still get why authors have not even provided a basic table with participant´s features.
Discussion
-It is fair to comment that such observed vulnerable participants might worsen their situation during the Covid-19 pandemic, since sexual habits have been observed to change over this period, including mental health derived from such sexual habits.
https://www.nature.com/articles/s41443-021-00494-9
https://pubmed.ncbi.nlm.nih.gov/35010405/
-Limitations should comprise selection and recall bias.
Round 2
Reviewer 2 Report
While the authors have addressed some of my original concerns, and the paper is improved and reads better, I still have two concerns: the justification to include a male case is not convincing and raises some methodological questions, and a conceptual and/or “significance” argument for the original contribution of this research remains weak.
Author Response
Thank you to the reviewer for further consideration. We are pleased that he/she believes the paper has improved. On the two remaining issues:
Regarding the inclusion of one man - we have left this in as noted before because we feel it gives an opportunity to highlight the challenges of recruiting men in this area.
Regarding the justification for the study, we had strengthened this and believe we have provided the detailed context for the current work. Given that this was not raised by any other reviewer or the editor we have not made further changes. The reviewer did not give specific recommendations to change this further.
Reviewer 4 Report
The authors addressed well my queries. Indeed, it is a qualitative study, but that does not mean that it can´t be improved or quantitative perspectives or analyses added.
Author Response
This review does not include any specific recommendations to be addressed. We acknowledge the point the referee makes regarding a quantitative perspective but did not feel it required further amendment. We thank the reviewer for this further review.